# Primary Health Care System Strengthening Project in Sri Lanka: Status and Challenges with Human Resources, Information Systems, Drugs and Laboratory Services

**DOI:** 10.3390/healthcare10112251

**Published:** 2022-11-10

**Authors:** Pruthu Thekkur, Manoj Fernando, Divya Nair, Ajay M. V. Kumar, Srinath Satyanarayana, Nadeeka Chandraratne, Amila Chandrasiri, Deepika Eranjanie Attygalle, Hideki Higashi, Jayasundara Bandara, Selma Dar Berger, Anthony D. Harries

**Affiliations:** 1Centre for Operational Research, International Union Against Tuberculosis and Lung Disease (The Union), 75001 Paris, France; 2Department of Health Promotion, Rajarata University of Sri Lanka, Mihintale, Anuradhapura 50300, Sri Lanka; 3The Union-South East Asia (USEA) Office, New Delhi 110016, India; 4Yenepoya Medical College, Yenepoya (Deemed to be University), Mangalore 575018, India; 5The Foundation for Health Promotion, 21/1 Kahawita Road, Dehiwala 10350, Sri Lanka; 6Department of Community Medicine, Faculty of Medicine, University of Colombo, Colombo 00300, Sri Lanka; 7The World Bank, Colombo 00300, Sri Lanka; 8Project Management Unit, Primary Health Care System Strengthening Project (PSSP), Colombo 00300, Sri Lanka; 9Department of Clinical Research, Faculty of Infectious and Tropical Diseases, London School of Hygiene and Tropical Medicine, London WC1E 7HT, UK

**Keywords:** noncommunicable diseases, package of essential NCD interventions, primary healthcare, health system, universal health coverage, operational research

## Abstract

A Primary Healthcare-System-Strengthening Project (PSSP) is implemented by the Ministry of Health, Sri Lanka, with funding support from the World Bank for providing quality care through primary medical care institutions (PMCIs). We used an explanatory mixed-methods study to assess progress and challenges in human resources, drug availability, laboratory services and the health management information system (HMIS) at PMCIs. We conducted a checklist-based assessment followed by in-depth interviews of healthcare workers in one PMCI each in all nine provinces. All PMCIs had medical/nursing officers, but data entry operators (44%) and laboratory technicians (33%) were mostly not available. Existing staff were assigned additional responsibilities in PSSP, decreasing their motivation and efficiency. While 11/18 (61%) essential drugs were available in all PMCIs, buffer stocks were not maintained in >50% due to poor supply chain management and storage infrastructure. Only 6/14 (43%) essential laboratory investigations were available in >50% of PMCIs, non-availability was due to shortages of reagents/consumables and lack of sample collection–transportation system. The HMIS was installed in PMCIs but its usage was sub-optimal due to perceived lack of utility, few trained operators and poor internet connectivity. The PSSP needs to address these bottlenecks as a priority to ensure sustainability and successful scale-up.

## 1. Introduction

Noncommunicable diseases (NCDs) are the leading cause of morbidity and mortality, accounting for 74% of global deaths, of which more than one-third were ‘premature’ deaths [1,2,3]. NCDs alone are estimated to cause an economic loss of about USD 47 trillion between 2011 and 2030 due to disability and premature deaths [4]. NCDs pose additional challenges to healthcare systems in low- and middle-income countries (LMICs) that are still battling communicable diseases, maternal, neonatal and nutritional disorders [5]. 

The World Health Organization’s (WHO) Global Action Plan for the prevention and control of NCDs (2013–2020) aims to achieve a 25% reduction in ‘premature’ NCD mortality by 2025 [6]. The Sustainable Development Goals (2015) and the United Nations High-Level meeting on NCDs (2018) suggested universal health coverage packages, including prevention, diagnosis and management of NCDs, for achieving the global targets of reducing ‘premature’ deaths [7]. This requires a strengthening of primary healthcare (PHC) with infrastructure, trained human resources, health management information systems (HMISs) and an uninterrupted availability of drugs and laboratory investigations for quality NCD care provision [8,9].

To guide the integration of NCD care with PHC, the WHO published a “Package of Essential NCD interventions for primary care in low-resource settings” (WHO PEN) in 2010, which provided a prioritized set of cost-effective interventions suitable for LMICs [10]. However, studies from LMICs assessing the integration of NCD care in PHC have shown critical gaps in trained manpower, drug availability, laboratory services and HMISs [11,12,13,14,15,16,17,18]. The reasons for such gaps are generally context specific, requiring detailed enquiry for action [19,20]. 

In Sri Lanka, an LMIC in the South Asia region, NCDs account for 81% of total deaths and 77% of disability-adjusted life years (DALYs) [21]. However, NCD care provision has remained suboptimal as existing primary medical care institutions (PMCIs) in the country have had a low capacity to provide optimal curative care services [22,23]. The Ministry of Health (MoH) constituted a Technical Expert Committee in 2017 to provide policy directions for reorganising PHC for provision of NCD care in PMCIs [24,25,26]. The committee advised that PHC be reorganised with improved infrastructure and processes to attain the goals of universal health coverage with a focus on NCD care. 

Accordingly, the MoH initiated the ‘Primary Healthcare-System-Strengthening Project (PSSP)’ in 2018, with technical and financial support from the World Bank [27]. The project focuses on three thematic areas: first, reorganisation of the PHC by defining the catchment area and population for each PMCI (empanelment); second, strengthening the PMCIs with trained manpower, optimizing drug supply chain management systems and expanding laboratory service capacity; and third, establishing a technology-based HMIS to provide electronic personal health records (PHRs) [28,29]. By 2023, the PSSP plans to strengthen 550 of the 990 PMCIs in the country [29].

In the first phase of PSSP (2019), 63 selected PMCIs across the nine provinces of the country were strengthened. However, there was no prior systematic assessment as to whether the PMCIs under PSSP had attained expected standards in terms of trained manpower, drug availability, laboratory services and HMIS. Such an assessment, using a combination of quantitative and qualitative techniques, could provide an in-depth understanding of operational realities and help programme managers to address bottlenecks in reorganisation [15,16,19,30]. The insights from the early experiences of strengthening PMCIs would also enable the MoH and the provincial health authorities, jointly with the project management unit (PMU), to optimise the PSSP implementation in the PMCIs in future. In 2021, we, therefore, conducted a study in nine selected PMCIs to assess if they were reorganised according to the standards endorsed by the MoH and to explore the challenges perceived by the healthcare workers (HCWs) implementing this project.

## 2. Materials and Methods

### 2.1. Study Design

This was an explanatory mixed-methods study [30]. The quantitative component was a cross-sectional descriptive study using a self-designed checklist. This was followed by the qualitative component, which was a descriptive study with in-depth interviews among healthcare workers.

### 2.2. Study Setting

#### 2.2.1. General Setting

Sri Lanka is an island country with a population of 21.8 million in 2019 [31]. Administratively, the country is divided into nine provinces, each governed by an autonomous provincial council [32]. The provinces are further divided into districts, administered by a district secretariat. The most peripheral local administrative units are Grama Niladhari (GN) divisions [32]. 

#### 2.2.2. Specific Setting

##### Healthcare System in Sri Lanka

The health system in Sri Lanka is a mix of both public and private care providers (Figure 1). The public-health system provides both preventive and curative services free of cost. The former focuses mainly on maternal and child health services delivered through divisional health units known as Medical Officer of Health (MOH) areas with a catchment population of 50,000 to 100,000 [33]. Curative services are divided into three levels: primary, secondary and tertiary. Patients can avail services at any level of care and need not necessarily move sequentially from primary to tertiary level [24]. Primary medical care institutions (PMCIs), which are the focus of PSSP, include primary medical care units (PMCUs) and divisional hospitals (DHs). The PMCUs provide only outpatient services while DHs provide inpatient and delivery services as well [33]. The PMCIs are governed by provincial health departments [34].

##### PSSP Project

The PSSP is managed and monitored by a PMU established within the MoH. The personnel in the PMU includes full-time staff deputed from MoH and the State Ministry of Provincial Councils and Local Government Affairs, along with a few others employed on a contractual basis. The PSSP receives financial aid from the World Bank, dependent on producing required results against nine agreed Disbursement-Linked Indicators (DLIs) on an annual basis [28]. The PSSP project was initiated in 63 PMCIs (4 PMCUs and 59 DHs) in the first year (June 2019) and the key activities are described below.

##### Empanelment of the Population

Process of empanelment was limited to the primary preventive healthcare institutions in Sri Lanka. With the PSSP, this concept was expended to the PMCIs in 2018. Empanelment refers to the process of identifying, assigning, actively reviewing and updating people’s health data. The PMCI is primarily responsible for the empanelment of residents of the GN division assigned to it by registering them, providing them unique personal health numbers (PHN) and creating their PHR. Once created, the PHR can be accessed and updated at any public-health facility.

##### Trained Human Resources

In accordance with guidelines of MoH, a minimum of two medical officers, one nursing officer and a dental surgeon constitute a core team and were supposed to be present in all the PMCIs [27]. Additionally, a public-health nursing officer (PHNO), health promotion officer, medical laboratory technician (MLT) and data entry operators (DEO) have been proposed to facilitate PSSP activities. These HCWs are expected to be trained on guidelines for screening and management of major NCDs, the process of empanelment and other aspects of PSSP.

##### Essential Drugs and Drug Supply Chain Management

The PMCIs are supposed to ensure availability of all essential drugs in blister packs and dispense drugs for a period of one month for patients with NCDs. A list of 18 essential drugs for PMCI was prepared in accordance with the guidelines of MoH (Appendix A). The pharmacist or dispenser forecasts requirements based on the prevailing morbidity and drug utilization patterns in the previous year and indents drugs online through the medical supplies management information system (MSMIS). The drugs indented are expected to be delivered within a month. Adequate space and optimum storage conditions for maintaining a three-month stock of drugs are supposed to be present in each PMCI. A detailed inventory and supply chain register is maintained in paper-based registers currently at the PMCI.

##### Laboratory Services

The PSSP has developed a basic investigation package that needs to be made available through all PMCIs (Appendix A). The project supports the establishment of a functional laboratory service network with apex laboratories in selected DH and secondary hospitals. All PMCIs are supposed to have certain point-of-care tests (POCT), such as blood glucose and serum cholesterol. For essential investigations not available at a PMCI, the providers are supposed to establish a system/network for sample collection and transportation (SCT) to apex laboratories and retuning the investigation reports back to the respective PMCIs.

##### Health Management Information System (HMIS)

All the PMCIs are supposed to have an online web-based HMIS platform managed by the HMIS unit of the MoH and used for the empanelment process. The said system has to support the information management for NCD surveillance at healthy lifestyle centres (HLCs). The health staff at the PMCI are supposed to register individuals from the assigned GN divisions by generating unique PHN on HMIS and creating PHRs. The clinical details are supposed to be updated on the HMIS during each health facility visit. The PHRs can be accessed on the HMIS using the PHN and updated across any health facility by trained DEOs, medical officers and nursing staff.

Paper-based PHRs (books) are handed over to registered individuals and they are expected to carry these while visiting any health facility, so that the management details can be documented. However, prior to the introduction of PHR books, the NCD patients had ‘clinic books’ in which follow-up details were documented.

### 2.3. Study Population

#### 2.3.1. Quantitative Component

The assessments were conducted in nine selected PMCIs, one from each of the nine provinces of Sri Lanka. Using a stratified random sampling method, six rural and three urban PMCIs were selected to represent the population distribution of the country. All the nine PMCIs included in the study were selected for implementation of PSSP by the PMU in June 2019. In eight of the PMCIs, implementation started immediately but in one it started in February 2020. Therefore, by the time of assessment, all the PMCIs had at least one year of implementation of PSSP.

#### 2.3.2. Qualitative Component

Purposive (extreme variation) sampling was used to select the facilities based on performance according to the quantitative assessment. The healthcare providers of the best and worst performing urban PMCIs and of the best and poor performing rural PMCIs were included. In total, we conducted 18 in-depth interviews among medical officers (4), nursing officers (3), drug dispensers/pharmacists (3), DEO (1) and programme managers at the provincial/district level (7). Sample size was guided by saturation of findings.

### 2.4. Data Collection, Study Variables, Data Source and Study Tools

#### 2.4.1. Quantitative Component

Trained research staff completed a pretested facility assessment checklist (Appendix A) through non-participatory observation and review of records maintained at the PMCI. The checklist included details of trained human resources, laboratory services, essential drugs and supply chain and HMIS.

#### 2.4.2. Qualitative Component

In-depth interviews were used as a method of qualitative data collection. In-depth interviews are a form of semi-structured interview conducted to gather information from key informants who have personal experiences, attitudes, perceptions and beliefs related to the topic of interest [35]. Typically, the interviewer is guided by an interview guide which lists the key issues to be explored. However, the sequencing and wording of the questions are modified based on the responses of the participant and flow of conversation. The interview guides used to explore challenges in human resources, laboratory services, essential drugs and supply chain and HMIS are provided in Appendix A.

In-depth interviews were conducted by research consultants, who were medical doctors familiar with the health system (one male and one female), fluent in the local languages (Sinhala and Tamil) and experienced in qualitative research. The interviews were conducted approximately ten days after the checklist-based assessment and the findings of the quantitative study were used to contextualise the respondent-specific interview guide for the qualitative exploration. The interviews were audio recorded and were used to prepare the transcripts. The average duration of the interviews was 32 min (range 8 to 80).

### 2.5. Data Analysis

#### 2.5.1. Quantitative Component

The data collected on the checklist were entered using an EpiCollect5 application (Wellcome Sanger Institute, Cambridge, UK), a mobile-phone-based data-capture tool. The data were downloaded and analysed using Stata version 12 (StataCrop LP, College Station, TX, USA). Frequencies and percentages were used to summarise the total number of PMCIs with available adequately trained manpower, essential drugs and MSMIS, buffer stocks of essential drugs, laboratory/diagnostic services and functional HMIS.

#### 2.5.2. Qualitative Component

The transcripts were prepared in English on the same day of interview. Thematic analysis was performed by the PI (PTK) using Atlas-Ti software to identify themes on the challenges in strengthening and reorganisation of PMCIs. The second investigator (AMVK) reviewed the analysis and decisions on coding/categories and theme generation were decided via consensus. Similar codes were combined into categories to describe certain themes. To ensure that the results reflected the data, the codes/categories were related back to the original data. The findings were reported as per ‘Consolidated Criteria for Reporting Qualitative Research’ guidelines [36].

## 3. Results

Of the nine PMCIs selected, three each were DH-A and DH-C, two were PMCUs and one was DH-B. Seven (DH) PMCIs functioned 24 h a day, while the rest (PMCUs) functioned for only 8 hours during the day. Outpatient attendance numbers in the month preceding data collection ranged from 50 to 196.

### 3.1. Human Resource and Training

Checklist-based assessments showed that a minimum of two medical officers and one nursing officer was available in all the nine PMCIs. Among additional proposed staff, DEOs were present in four (44%) PMCIs, MLTs in three (33%) and PHNO in two (22%). There was no information regarding the total number of ‘approved positions’, other than for medical officers and nursing officers. At least one medical officer trained in empanelment and NCD guidelines was present in eight (89%) PMCIs and at least one trained nurse was available in seven PMCIs (78%). Of the 38 medical officers and 82 nursing officers available in the nine PMCIs, 19 (50%) and 22 (27%) were trained, respectively.

In total, 46 codes related to challenges with human resources and training were identified from the transcripts of HCW interviews, which were reduced to 11 categories describing the challenges and their impact on the HCWs. The categories highlighted in Figure 2 are explained in detail with relevant verbatim quotes in Appendix A.

There was reluctance among HCWs to work in the rural PMCIs due to their preference for urban neighbourhoods and lack of recognition for their hard work. The HCWs felt that annual transfers and on-demand transfers of staff and the temporary recruitment of DEOs led to frequent turnover of trained staff. The HCWs felt that the PMCIs were usually understaffed, especially with respect to PHNOs, MLTs and DEOs. With a lack of PHNOs for carrying out community activities, the DEO for online registration and the MLT for sample collection and processing, these responsibilities were placed on the existing HCWs (medical and nursing officers), leading to exhaustion and work stress.

These issues were compounded by deficiencies in training in the newer activities introduced after PSSP. Without adequate funds and experienced resource persons, the training was not frequently conducted, especially with the onset of the COVID-19 pandemic. Even when conducted, some HCWs were unable to participate in the training due to a lack of spare staff to conduct their work at the PMCI and restrictions from supervisors. The HCWs felt there was no standardised training content, with various programme units supporting PSSP activities delivering different messages, and this led to confusion among trainees. Further, those who were trained did not make much effort to train the rest of the staff in the PMCIs.

Despite these challenges, the HCWs appreciated the constant push and monetary support from the PSSP to recruit DEOs. They felt that the PMCIs under the PSSP received preference during allocation of staff from the common pool because of efforts from the PMU of PSSP. Some HCWs felt that there were substantial efforts to train them on empanelment and registration. Further, the well-performing PMCIs acted as a resource centre for training the HCWs from the second set of PMCIs included under the project.

### 3.2. Essential Drugs and Supply Chain Management

During the assessment, 11 (61%) of the 18 essential drugs were available in all the PMCIs. Salbutamol respiratory solution, Enalapril maleate, Gliclazide and Theophylline tablets were not available in one of the PMCIs. More than 50% of the PMCIs had less than three months of buffer stock for 12 of the 18 drugs. Stock outs for Gliclazide and Enalapril tablets were reported in three (33%) PMCIs during 2020. During the previous indent, all except two PMCIs received the drugs within one month of indenting (Table 1).

Eight (89%) PMCIs disbursed drugs for more than one month but none in blister packs. Adequate drug storage space was available in seven (78%) PMCIs, with the majority (67%) not having air conditioning in the drug stores. Seven (78%) PMCIs had an online MSMIS for drug indenting. The drug indenting and dispensing registers in the PMCIs were incomplete in four PMCIs with dates of indenting and receipt of drugs missing.

In total, 32 codes related to challenges with essential drugs and supply chain management were deduced during analysis of interviews with HCWs. The codes were summarised under five categories (Figure 3, Appendix A).

The HCWs perceived a suboptimal drug supply chain management as one of the reasons for shortages of essential drugs. The pharmacists or dispensers were either not available or were not trained in timely drug forecasting and ordering. The MSMIS platform designed for this purpose was not functional due to a lack of trained personnel to handle it. Once ordered, delivery of drugs was often delayed due to a lack of dedicated vehicles for transporting drugs to some of the PMCIs. The HCWs complained that insufficient drug storage space, lack of air conditioning and storage of drugs in transparent pill bottles contributed to suboptimal drug storage, leading to apprehensions regarding quality of drugs and inability to store adequate buffer stocks.

The HCWs reported unavailability of non-essential drugs, especially with an increase in demand from patients with specialised needs being referred to PMCIs for follow-up from secondary/tertiary hospitals. Such drugs were not easily available, as they could not be procured locally and had to go through a long process for procurement. Ultimately, stock outs in drugs meant that the patients had to purchase drugs from private pharmacies, incurring out-of-pocket expenditures. 

Another challenge related to drugs was non-availability of blister packs, resulting in dispensing of loose drugs in plastic covers, which could not be stored safely for a month and which were also perceived to be of low quality among the patients. This was also reported as a challenge for inventory management, as counting individual tablets was a tedious and error-prone exercise. 

The HCWs felt there had been a positive change in availability of the essential drugs since the introduction of PSSP. Under the PSSP, PMCIs had to prioritise the supply of drugs and there was less chance of a stock-out situation. Similarly, there was less chance of having drugs expire in the PMCIs, as there was an increase in service utilization. Though there were lacunae in training and establishment of MSMIS, the HCWs perceived that MSMIS brought efficiency in supply chain management.

### 3.3. Laboratory Services

In-house laboratory facilities were present in seven (78%) PMCIs. All PMCIs had functional glucometers and seven (78%) had functional cholesterol meters. Out of 14 essential investigations reported, 6 (43%) were available in more than 50% of the PMCIs. Blood glucose and total cholesterol estimations were available through all the nine PMCIs (seven in-house and two in the apex laboratory). Serum creatinine estimation was available through seven (78%) PMCIs (four in-house and three in the apex laboratory). In one of the PMCIs utilizing services from an apex laboratory, SCT was not established (Table 2). In other places, the staff collected blood and patients dropped their samples at the apex laboratory. For tests that were not available through the PMCI, patients had to approach private laboratories.

In total, 25 codes related to challenges with laboratory services in PMCIs were deduced from interviews with HCWs, which were reduced to 11 categories (Figure 4, Appendix A).

The HCWs felt that some of the reasons for non-availability of investigations in the PMCI were lack of laboratory space, poor maintenance of the auto analyser and lack of reagents and consumables for POCTs. Though SCT to apex laboratories for investigations unavailable in PMCI had been suggested, the HCWs noted that the system was not well established due to a lack of programmatic support. There were no dedicated personnel for SCT from PMCI to apex laboratories. The turnaround time for investigations through SCT was generally high and these delays increased further during COVID-19 times. 

The HCWs felt that the shortage of trained MLTs was a major challenge in ensuring laboratory investigations through PMCIs. Cancellations of certain investigations were frequent in apex laboratories due to a lack of trained MLTs and shortage of reagents. Investigations in private laboratories were unaffordable for most patients. As private laboratories were not stringently regulated, there were concerns about the quality of their services.

The HCWs felt that with the PSSP project, there was increased availability of POCT devices, consumables and their utilization in the PMCIs. Though there were deficiencies, the HCWs were appreciative of the SCT system established after PSSP, which had enabled them to access the follow-up investigations for better management of their NCD patients.

### 3.4. Health Management Information System (HMIS)

All the PMCIs had at least one computer with internet connection. The HMIS was available in all the PMCIs and was used for registering individuals and issuing PHN (empanelment). All PMCIs had hard copies of the PHR books. However, the PHR books had not been issued to all registered individuals in seven (78%) PMCIs. Electronic PHR was available in six (67%) PMCIs but only three (33%) updated clinical details in it during follow-up visits. In other PMCIs, follow-up details were documented in either ‘clinic books’ or PHR books.

In total, 25 codes related to challenges in using HMIS were deduced from the transcripts of interviews with HCWs, which were reduced to 13 categories (Figure 5, Appendix A).

Personnel-related issues: HCWs across all cadres complained about shortage of DEOs, as this required either the nursing officer or the doctor to enter data in the HMIS. They regarded data entry as an additional task with no relevance to routine care provision. The programme managers also mentioned that systematic collection and digitization of data was not perceived to be useful by HCWs. These challenges resulted in poor utilization of HMIS, even though the required equipment had been provided. 

Infrastructural issues: The HCWs complained that the HMIS module and PHR books were not available in the initial days of project and, therefore, PHN and PHR books could not be issued to all the registered individuals. They felt that the current supply of HMIS-enabled laptops (one or two laptops per PMCI) was not sufficient for simultaneous data entry from multiple service delivery points. Although internet connection was provided, poor connectivity made the use of the online HMIS portal for real-time registration impossible, especially during outreach clinics. 

The HCWs mentioned that the online HMIS platform crashed frequently, especially during working hours when multiple people used it simultaneously. The HMIS programme attributed this deficiency to allocation of low server space (16 GB instead of 64 GB RAM) from the government cloud. The HCWs perceived lack of an interactive dashboard for summary reports as a major challenge, which prevented them from monitoring their performance. They also mentioned that it was difficult to locate digital PHRs on HMIS because manual entry of the 10-character alphanumeric PHN (identifier) for tracing was prone to transcription errors.

The programme managers mentioned that HCWs in some of the PMCIs duplicated their efforts by documenting demographic details in the paper-based register first and then entering these data into online HMIS for generating PHN. The HCWs might have missed entering the data into HMIS in a timely manner, as there were no standard timelines for transcribing data. The HCWs complained that, despite issuing the PHR books, the patients failed to bring them during follow-up visits. Thus, the PHR books and electronic PHR remained incomplete, defeating the purpose of issuing the PHR. 

Notwithstanding the challenges, HCWs appreciated both the efforts by the PSSP to establish the HMIS and initiatives taken by programme managers for effectively communicating changes in the module.

## 4. Discussion

This was the first assessment on how well PSSP had worked in strengthening the PMCIs and what were the challenges in ensuring that trained manpower, functional HMIS with intra- and inter-operability of services within the PMCIs and between different levels; uninterrupted availability of drugs and investigations was provided for quality NCD care. Although commendable progress has been made, there were several challenges which require urgent attention if the standards defined by the MoH or PSSP are to be met.

The study had several strengths. First, the mixed-methods design helped in elucidating and understanding the real-world situation of PMCI reorganisation, gaps and challenges in reorganisation. Second, the checklists and interview schedules used for facility assessments were validated in consultation with the programme managers and were amended before use. Third, all the research staff were trained on how to fill in the checklist-based assessment and dummy data extraction was practiced during the training, minimizing inter-observer bias. Finally, the quality of the completed checklists was reviewed by the PI and research consultants on a daily basis and was re-validated during in-depth interviews at the PMCI. 

The study had a few limitations. First, only nine PMCIs were included in the quantitative assessment due to financial constraints and the COVID-19 situation and they, therefore, may not be representative of all the PMCIs. However, efforts were made to select one PMCI each from the nine provinces to accommodate variation across the provinces. Second, the non-availability of baseline (prior to implementation of PSSP) data of the PMCIs selected in the study limited our ability to quantitatively assess the impact of PSSP on human resource, HMIS, laboratory services and drug availability. Third, records and registers were not well maintained at the PMCIs with lots of missing data, making it difficult to extract some of the data from a few PMCIs. Fourth, the PMU staff were not interviewed to understand the challenges at the central level for reorganising the PMCIs. 

All the PMCIs had the minimum required number of medical and nursing officers. The implementation of PSSP brought in additional work, but PHNOs, DEOs and MLTs needed for executing this work were not available in most of the PMCIs. Therefore, the additional tasks of PSSP were shifted to existing staff, who felt overburdened and stressed. Although ‘task shifting’ is recommended as an important intervention in LMICs, if it is implemented irrationally, it can lead to inefficiency [37]. The PMCIs did not have documents on the number of ‘approved’ positions for cadres other than medical and nursing officers, highlighting the need for developing and conducting staffing establishment assessments for PMCI based on service utilization and size of the catchment areas [38,39,40]. 

The non-availability of DEOs was largely due to high turnover because of the contractual nature of the DEO position. Studies from other LMICs also reported similar inefficiencies with contractual arrangements for the health workforce [41,42]. We recommend immediate deployment of DEOs and creation of a permanent cadre of DEOs in future for maintaining and updating the PHR. Like other countries, the shortage of MLTs in PMCIs was due to a lack of MLT training programs in Sri Lanka [43,44]. Therefore, training of MLTs through bachelor programs or short-term training programs has to be prioritized [45,46]. As a stop-gap measure, one of the existing nursing officers can be deployed as PHNO after being relieved from curative care services and trained in community mobilisation. 

Pre-implementation training on PSSP was conducted for key personnel from the PMCIs. The training coverage was sub-optimal due to poor attendance and inability to ensure “trickle-down” training of the remaining personnel in PMCI. The training was not effective due to a lack of standardised training materials and insufficient emphasis on practical issues. Comprehensive training modules need to be developed by the PMU in consultation with major stakeholders, such as the NCD programme unit, planning unit and directorate of health services. Involvement of pro-active HCWs with experience in implementing PSSP in the first phase and use of case studies on early implementation experiences could help to orient trainees on practical aspects. Online training modules and short video lectures could be developed to serve as ready references and to enable the conduct of training during situations, such as the COVID-19 pandemic. 

Blood glucose and blood cholesterol estimations with POCTs were generally available, which is appreciable given the utility of the POCTs in NCD care provision [47,48,49,50]. However, there were instances of shortages of consumables (strips), leading to underutilization of POCTs, as reported from Ghana [51]. Unavailability of other essential investigations was largely due to shortages of reagents and MLTs. Many samples had to be transported to the apex laboratories but the SCT system performed sub-optimally, mimicking the operational challenges in SCT reported elsewhere [52,53,54,55]. 

The laboratory services need to be strengthened as a priority, either by improving the availability of essential investigations at the PMCI or by establishing robust SCT to ensure quality NCD care. The reagents and consumables have to be supplied in sufficient quantities and in an uninterrupted manner based on previous utilization statistics. There is a need for further research to find better models to improve the efficiency of SCT, so that there is timely availability of investigations and reduced turnaround times for results [56,57,58,59]. 

More than 50% of the drugs on the essential drug list were available at the PMCIs. However, some PMCIs had instances of stock outs of essential drugs. The stock outs were largely attributed to inefficiencies in drug forecasting and use of MSMIS for indenting, which are in line with similar findings that have been reported from other countries [60,61]. The drugs were not available in blister packs and were disbursed as loose drugs, leading to poor perception on the quality of drugs in the PMCI [62]. There was limited storage space, poor maintenance of drug registers and no air conditioning in most sites. Furthermore, the drugs required for patients back-referred from the secondary or tertiary hospitals were not available at the PMCIs and had to be purchased from private pharmacies.

There is a need for uninterrupted supplies of essential NCD drugs in order to reduce out-of-pocket expenditures. The pharmacists and dispensers need to be trained in drug forecasting, indenting using MSMIS, drug-storage practices and record maintenance in order to bring more efficiency to the supply chain management. Blister packs need to be introduced in order to overcome patients’ concerns about the quality of drugs. Physical infrastructure for safe long-term storage of drugs, including buffer stocks, needs to be strengthened. Moving forward, the essential drugs list of PMCIs could be rationalised, to accommodate some of the oral NCD drugs that are currently available only at secondary and tertiary hospitals.

An online HMIS was available in all the PMCIs. Similar to findings from other countries, the HMIS was not utilised optimally to manage PHRs due to poor internet connectivity, frequent computer crashes because of inadequate server space, glitches in some modules and other ill-defined processes [63,64,65]. Shortages of DEOs and reluctance of HCWs to use the HMIS, because they failed to appreciate the relevance of the HMIS data, led to underutilization of the system. 

As the HMIS is an online platform, there is an urgent need to improve the internet connectivity at the PMCIs. The server capacity of the HMIS should be augmented for smooth functioning, which becomes more relevant in the context of expansion of the PSSP to newer PMCIs. Moving forward, there is a need to develop an offline HMIS module to limit the dependence on continuous internet connectivity, especially in remote areas. Development of dashboards, which can aid HCWs in monitoring their progress and developing their work plans, would add significant value to the current HMIS.

## 5. Conclusions

This study conducted among PMCIs, strengthened through the first phase of PSSP, showed that significant efforts have been made in improving population screening for NCDs at PMCIs/HLCs, the availability of trained manpower, essential drugs, laboratory services and HMIS at HLCs and PMCIs. However, there were several facility-specific challenges that need to be addressed as a priority to ensure sustainability and successful scale-up of PSSP.

## Figures and Tables

**Figure 1 healthcare-10-02251-f001:**
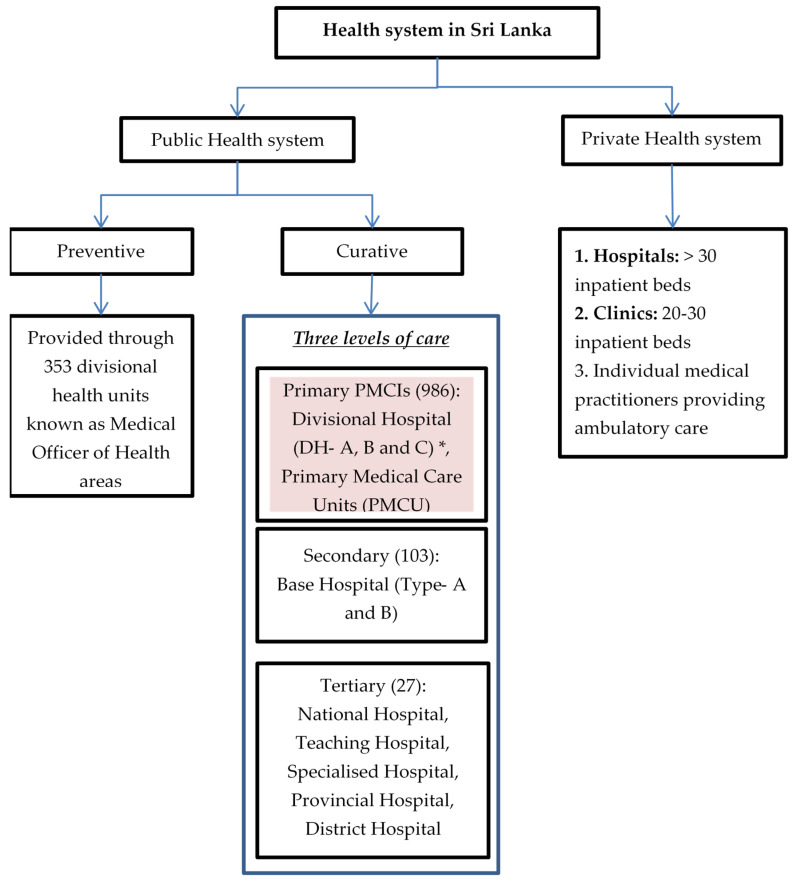
Health system in Sri Lanka, 2021. * DH-A has >100 inpatient beds, DH-B has 50–100 inpatient beds and DH-C has <50 beds.

**Figure 2 healthcare-10-02251-f002:**
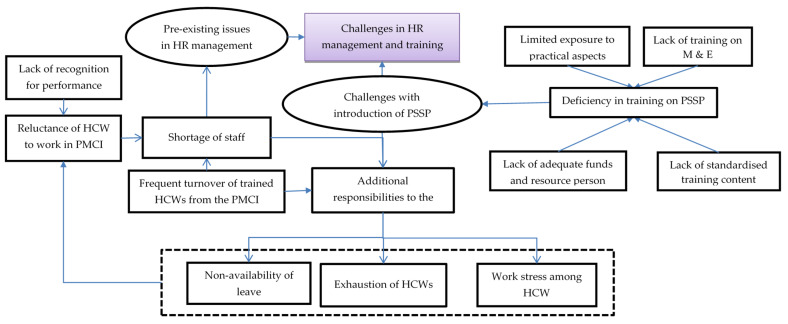
Challenges with human resource management and training in PMCIs supported by PSSP in Sri Lanka, 2021. Abbreviation: HCWs: Healthcare Workers; PMCI: Primary Medical Care Institution; PSSP: Primary Healthcare-System-Strengthening Project (PSSP); M and E: Monitoring and Evaluation.

**Figure 3 healthcare-10-02251-f003:**
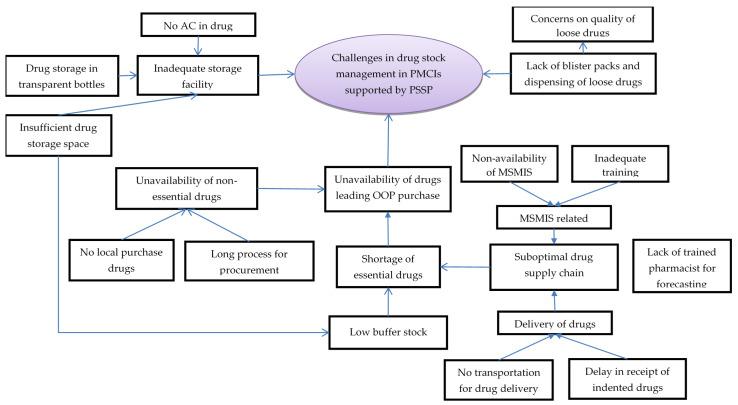
Challenges in drug stock management in PMCIs supported by PSSP in Sri Lanka, 2021. Abbreviation: PMCI: Primary Medical Care Institution; PSSP: Primary Healthcare-System-Strengthening Project (PSSP); AC: Air Conditioner; MSMIS: Medical Supplies Management System.

**Figure 4 healthcare-10-02251-f004:**
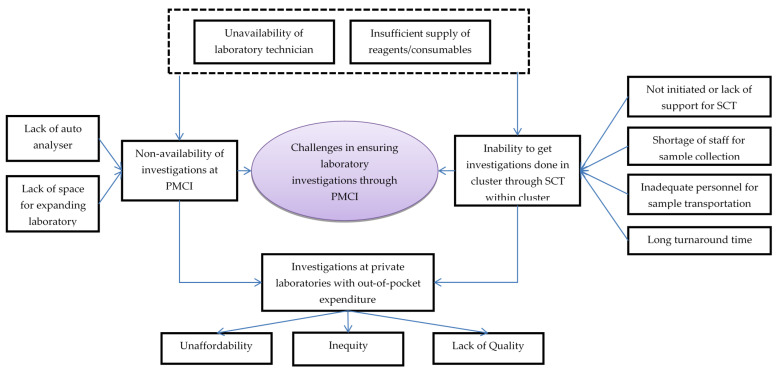
Challenges in laboratory investigations conducted through PMCI supported by PSSP in Sri Lanka, 2021. Abbreviation: PMCI: Primary Medical Care Institution; PSSP: Primary Healthcare-System-Strengthening Project (PSSP); SCT: Sample collection and transportation.

**Figure 5 healthcare-10-02251-f005:**
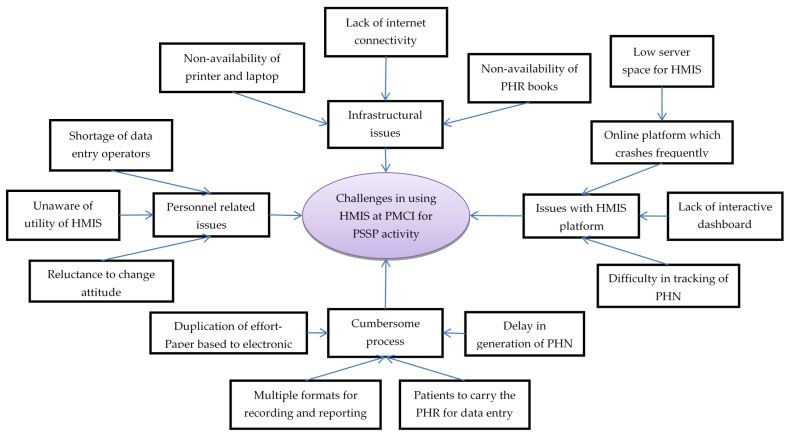
Challenges in using Health Management Information System established in PMCI supported by PSSP in Sri Lanka, 2021. Abbreviation: PMCI: Primary Medical Care Institution; PSSP: Primary Healthcare-System-Strengthening Project (PSSP); PHN: Personal Health Number; PHR: Personal Health Record; HMIS: Health Management Information System.

**Table 1 healthcare-10-02251-t001:** Availability of essential drugs, stock outs and receipt of drugs within a month of indenting in the nine selected PMCIs supported by the Primary Healthcare-System-Strengthening Project (PSSP) in Sri Lanka in 2021.

Name of the Essential Drug	Drugs Available, n (%)	Adequate Buffer Stock *, n (%)	Stock Outs in 2020, n (%)	Receipt of Drugs within a Month, n (%)
*Cardiovascular medicines*				
Anti-arrhythmic medicines				
Adrenaline tartrate (0.1%)—1 mL Ampoule	9 (100.0)	1 (11.1)	0 (0.0)	6 (85.7) a
Anti-thrombotic medicines				
Aspirin tablet 150 mg	5 (55.6)	1 (20.0) *	1 (11.1)	5 (100.0) b
Aspirin tablet 75 mg	3 (33.3)	3 (100.0) #	1 (11.1)	3 (100.0) #
Lipid-lowering agents				
Atorvastatin tablet 10 mg	9 (100.0)	8 (88.9)	1 (11.1)	8 (88.9)
Anti-anginal medicines				
Glyceryl trinitrate tablet 0.5 mg sublingual	9 (100.0)	5 (55.5)	1 (11.1)	7 (77.8)
Nifedipine slow release tablet 20 mg	9 (100.0)	7 (77.8)	2 (22.2)	7 (87.5) c
Anti-hypertensive drugs				
Atenolol tablet 50 mg	9 (100.0)	6 (66.7)	1 (11.1)	7 (77.8)
Enalapril maleate tablet 5 mg	8 (88.9)	7 (77.8)	3 (33.3)	8 (88.9)
Frusemide—Injection 20 mg in 2 mL Ampoule	9 (100.0)	2 (22.2)	0 (0.0)	8 (88.9)
Frusemide tablet 40 mg	9 (100.0)	5 (55.5)	1 (11.1)	8 (88.9)
Hydrochlorothiazide tablet 25 mg	9 (100.0)	8 (88.9)	1 (11.1)	8 (88.9)
Losartan tablet 50 mg	9 (100.0)	7 (77.8)	1 (11.1)	9 (100.0)
*Medicine for diabetics (Oral hypoglycaemic)*				
Gliclazide tablet 40 mg and 80 mg	8 (88.9)	8 (100.0)	3 (33.3)	8 (88.9)
Metformin tablet 500 mg	9 (100.0)	9 (100.0)	2 (22.2)	8 (88.9)
*Anti-asthmatic medicines*				
Salbutamol Respiratory solution 0.5%/10 mL	8 (88.9)	6 (66.7)	0 (0.0)	7 (87.5) d
Salbutamol 2 mg tablet	9 (100.0)	3 (33.3)	0 (0.0)	8 (88.9)
Salbutamol 4 mg tablet	0 (0.0)	NA ‡	0 (0.0)	NA ‡
Theophylline Slow released tablet 125 mg	8 (88.9)	4 (44.4)	1 (11.1)	8 (88.9)

Note: We used the WHO Model list of Essential Medicines to categorize the essential drugs. Percentages are calculated with nine PMCIs as a denominator; * Aspirin 150 mg was not indented in four PMCIs, # Aspirin 75 mg tablet was not indented in six out of nine PMCIs. None of the PHC indented Salbutamol 4mg tablet. Hence these PMCIs were not included in the respective analysis.; a—Drug indenting and receiving dates were not available for two PMCIs; b—Drug indenting and receiving dates were not available for four PMCIs; c, d—Drug indenting and receiving dates were not available for one PMCI; ‡—The buffer stock for three months (or one quarter) = (Total number of tablets disbursed in 2020/4). The required buffer stock for three months was compared against the closing stock in the last month to assess the number of PMCIs with less than three months of buffer stock.

**Table 2 healthcare-10-02251-t002:** Availability of laboratory/diagnostic services in the nine selected PMCIs supported by the PSSP in Sri Lanka in 2021.

Name Laboratory/Diagnostic Services	Available through PMCI n (%)	Available at the PMCI n (%)	Availed at Apex Laboratories in the Cluster n (%)	Sample Collection Facilities for Specimens Sent to Apex Laboratories (%) ^a^
Blood Glucose	9 (100.0)	7 (77.8)	2 (22.2)	1 (50.0)
HbA1C	0 (0.0)	0 (0.0)	0 (0.0)	-
Total Cholesterol	9 (100.0)	7 (77.8)	2 (22.2)	1 (50.0)
Lipid Profile	2 (22.2)	2 (100.0)	0 (0.0)	-
Serum Creatinine	7 (77.8)	4 (57.1)	3 (42.9)	2 (66.7)
Urine for glucose	6 (66.7)	6 (100.0)	0 (0.0)	-
Urine for ketone bodies	2 (22.2)	2 (100.0)	0 (0.0)	-
UACR	1 (11.1)	1 (100.0)	0 (0.0)	-
OGTT	6 (66.7)	5 (83.3)	1 (16.7)	1 (100.0)
PAP-Smear	1 (11.1)	0 (0.0)	1 (100)	1 (100.0)
ALT/AST	4 (44.4)	4 (100.0)	0 (0.0)	-
Complete Blood Count	8 (88.9)	6 (75.0)	2 (25.0)	2 (100.0)
AFB stain for TB	2 (22.2)	1 (50.0)	1 (50.0)	1 (100.0)
HIV	3 (33.3)	0 (0.0)	3 (100.0)	3 (100.0)

Abbreviations: PMCIs: Primary Medical Care Institutions; PSSP: Primary Healthcare-System-Strengthening Project; ALT/AST: Alanine aminotransferase/Aspartate aminotransferase; HIV—Human immunodeficiency virus infection; HbA1C—Haemoglobin A1C; UACR—Urine Albumin Creatinine Ratio; OGTT—Oral Glucose Tolerance Test. Footnote: ^a^—The number of facilities offered sample collection and transport facilities if the test was done at the public cluster. The denominator for this column is number of facilities that utilized public clusters for laboratory/diagnostic investigation.

## Data Availability

Requests to access these data should be sent to the corresponding author.

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
