# Peer review of "Primary Health Care System Strengthening Project in Sri Lanka: Status and Challenges with Human Resources, Information Systems, Drugs and Laboratory Services"

_healthcare, 2022, doi:10.3390/healthcare10112251_

Round 1

Reviewer 1 Report

I would like to thank the authors for their work and this manuscript.

The present study aims at evaluating if PMCIs in Sri Lanka fits MoH standards as intended in the PSSP of 2018.

I first would like to highlight the considerable work of the authors as they conducted a mixed-methods study, but the manuscript should be thoroughly revised before being considered for publication. At this time, I would reject this paper.  

The introduction section is well written and gives the reader all the elements for understanding this paper.

Materials and Methods:

Your quantitative study is cross-sectional but are every of the nine PMCIs were studied using the same baseline? For example, one can imagine that a facility that has transitioned recently will suffer less from a shortage of drugs and staff than a facility that transitioned long ago (or vice versa).

In section 2.4.2, please explain a little bit about what in-depth interviews are. In addition, if you used an interview guide, please give the reader the most important topics you addressed and provide the full guide in an appendix. 

Results:

Your quantitative results are weak, and I am not sure they are relevant to your paper.

Table 1 is not necessary. All the information could be given in the text, in particular, because almost all of the data in the "number of individuals" columns is missing.

Table 2 is not relevant as it stands. Rather than making a "shopping list", it would be more interesting to present the drugs by therapeutic class.

As in table 1, these "descriptive statistics" do not add much because they are not representative of all PMCIs, and the sample of 9 structures is too small.

Your qualitative results are very interesting as your analytical approach.

All in all, my main recommendation is:

Your paper would benefit from presenting only the qualitative study given the lack of interest of the quantitative results.

I recommend you to read Caffery and al. (Journal of Telemedicine and Telecare, 2017), which describe how to use mixed studies to answer the same research question.

In your case, understanding the challenges when implementing this kind of lofty project IS relevant enough for publication.

Discussion:

I disagree your statement about the “real-world situation of PMCI reorganization” across the Sri Lanka. There is no guarantee in your results that each PMCI is representative of the PMCIs in its region.

What you are pointing out as a strength of your study seems to be a limitation considering the risk of bias (considering your quantitative study).

If you consider redesigning your study to present only your qualitative results, I recommend you rewrite the entire discussion to stick to those results.

Author Response

Thank you for the review. Please find below the response for comments.

The present study aims at evaluating if PMCIs in Sri Lanka fits MoH standards as intended in the PSSP of 2018. I first would like to highlight the considerable work of the authors as they conducted a mixed-methods study, but the manuscript should be thoroughly revised before being considered for publication. At this time, I would reject this paper. 

Response: Thank you for the comment. We have now revised the manuscript in line with your suggestions highlighting the qualitative findings and not drawing much inference from the cross-sectional quantitative assessment of nine PMCIs.

The introduction section is well written and gives the reader all the elements for understanding this paper.

Response: Thank you for comment.

Materials and Methods:

Your quantitative study is cross-sectional but are every of the nine PMCIs were studied using the same baseline? For example, one can imagine that a facility that has transitioned recently will suffer less from a shortage of drugs and staff than a facility that transitioned long ago (or vice versa).

Response: Thank you for the observation. All the nine PMCIs included in the study were selected for implementation of PSSP by the Project Management Unit  in June 2019. In eight of the PMCIs implementation started immediately but in one it started several months later in February 2020. Therefore, by the time of assessment, all the PMCIs had at least one year of implementation of PSSP. Thus, we feel there was enough time for reorganisation of the facilities selected in the study irrespective of their baseline characteristics. We have now detailed this in the manuscript in line number 184-188.

Also, we feel that the non-availability of baseline (prior to implementation of PSSP) data of the PMCIs selected in the study has limited our ability to quantitatively assess the impact of implementation of PSSP. We have highlighted this as a limitation in the manuscript in line number 454-457.

In section 2.4.2, please explain a little bit about what in-depth interviews are. In addition, if you used an interview guide, please give the reader the most important topics you addressed and provide the full guide in an appendix.

Response: Thank you for the suggestion. As suggested we have now explained what in-depth interviews are in line number 203-211. We have also provided Annex-4 with all the interview guides used in the study.

Results:

Your quantitative results are weak, and I am not sure they are relevant to your paper.

Response: Thank you for the observation. We too agree that the quantitative results are weak as the assessment was conducted only in nine out of 63 PMCIs. However, we believe this is an integral part of the manuscript and prefer to retain the quantitative component in the manuscript with some modifications based on your subsequent comments. The reasons for this are as follows:

  1. These articles will be used for advocating for better implementation of PSSP among policy makers of the Ministry of Health in Sri Lanka. We feel the quantitative data will be useful to highlight the existing gaps and the qualitative data complements this by providing the reasons/challenges for such gaps. We have used an explanatory mixed-methods study design (as even described by Caffrey et al), where the findings of quantitative and qualitative are meant to be interpreted in conjunction with each other.
  2. Accepting the small sample size of the quantitative component, we have not tried to generate any inferential statistics like confidence intervals. We have just used the descriptive approach to highlight gaps in quantitative form.
  3. Reviewer 2 thought the quantitative part was interesting. He went on to say ‘The survey is well designed and the results seem to reflect the overall situation in Sri Lanka’.

Table 1 is not necessary. All the information could be given in the text, in particular, because almost all of the data in the "number of individuals" columns is missing.

Response: Thank you for the suggestion. We have now removed Table-1 from the main narrative.

Table 2 is not relevant as it stands. Rather than making a "shopping list", it would be more interesting to present the drugs by therapeutic class. As in table 1, these "descriptive statistics" do not add much because they are not representative of all PMCIs, and the sample of 9 structures is too small.

Response: Thank you for the suggestion. We have now categorised the essential drugs in line with the WHO Model list of Essential Medicines. Modifications made in Table-1 (previously Table-2). As highlighted earlier, we prefer to retain the quantitative findings in the manuscript.

Your qualitative results are very interesting as your analytical approach.

Response: Thank you for the appreciation.

 All in all, my main recommendation is:

Your paper would benefit from presenting only the qualitative study given the lack of interest of the quantitative results. I recommend you to read Caffery and al. (Journal of Telemedicine and Telecare, 2017), which describe how to use mixed studies to answer the same research question. In your case, understanding the challenges when implementing this kind of lofty project IS relevant enough for publication.

Response: Thank you for the suggestion. As highlighted earlier, as this is an explanatory mixed-methods study, the findings from either quantitative or qualitative cannot be interpreted in silos. As mentioned by Caffery et al, we have used quantitative methods to substantiate the fact that there are gaps and qualitative methods to understand the implementation challenges and reasons for such gaps. We strongly feel we have adhered to the six points recommended in GRAMMS checklist and the COREQ guidelines for reporting qualitative findings.

The reason for limiting to only nine PMCIs in the assessment has been detailed in the manuscript (line number 452). We agree that nine is a small sample. However, it constituted 14% of the total PMCIs selected for implementation of PSSP during June 2019. Based on your suggestion, we have removed Table-1.

Discussion:

I disagree your statement about the “real-world situation of PMCI reorganization” across the Sri Lanka. There is no guarantee in your results that each PMCI is representative of the PMCIs in its region. What you are pointing out as a strength of your study seems to be a limitation considering the risk of bias (considering your quantitative study).

Response: Thank you for the comment. The emphasis on the ‘real-world situation of PMCI reorganisation’ was to highlight the advantage of using an explanatory mixed-methods design in understanding the problem. We too agree that the quantitative results are not generalizable considering the small sample size. In this context, we have edited the statement on strength in line number 442-444. Also, we have explicitly mentioned the small sample size for the quantitative assessment as a limitation in line 451-453. 

If you consider redesigning your study to present only your qualitative results, I recommend you rewrite the entire discussion to stick to those results.

Response: Thank you for the suggestion. We prefer to retain this as a mixed-methods study. However, as you might have noted the discussion largely revolves around the qualitative findings and similar observations from other study settings.

Reviewer 2 Report

The authors evaluated the progress and challenges of Primary Health Care System Strengthening Project (PSSP) being conducted in Sri Lanka by assessing several key points, such as human resources and drug availability at primary medical care institutions (PMCIs). This survey seems meaningful for clarifying the issues to be addressed for further improving health care system in the country. The survey is well designed and the results seem to reflect the overall situation in Sri Lanka. Additional information may help the readers to understand the situation of PSSP implementation and issues to be addressed, if relevant information is available.

The description of quantitative goals of PSSP by the ministry will be helpful for readers to understand what degree it has been achieved so far.

Comparables, such as the situation in other countries and the previous situation in Sri Lanka, make it easier to understand the current level of PSSP achievement.

Many abbreviations are used in the manuscript. If the journal's editorial policy permits, description of the relationship between abbreviations and abbreviations in one place would be helpful for readers.

Author Response

Thank you for the review. Please find below the point-by-point response to the comments.

The authors evaluated the progress and challenges of Primary Health Care System Strengthening Project (PSSP) being conducted in Sri Lanka by assessing several key points, such as human resources and drug availability at primary medical care institutions (PMCIs). This survey seems meaningful for clarifying the issues to be addressed for further improving health care system in the country. The survey is well designed and the results seem to reflect the overall situation in Sri Lanka. Additional information may help the readers to understand the situation of PSSP implementation and issues to be addressed, if relevant information is available.

Response: Thank you for the appreciation.

The description of quantitative goals of PSSP by the ministry will be helpful for readers to understand what degree it has been achieved so far.

Response: Thank you for this comment. As described in the specific setting, the PSSP set a goal of ensuring all the infrastructure during reorganization. As these PMCIs had already been implementing activities under the project for more than a year, they were supposed to have all the infrastructure prior to assessment. The benchmarks are well described in references 27 and 28. We have now explicitly highlighted that all the PMCIs are supposed to have the recommended human resource establishment, the essential drugs, essential laboratory services and a functional HMIS (Line number 143, 149, 161 and 168).

Comparables, such as the situation in other countries and the previous situation in Sri Lanka, make it easier to understand the current level of PSSP achievement.

Response: Thank you for this comment. One of the problem with this assessment was that we did not have baseline data (prior to implementation of PSSP) on the infrastructure availability in Sri Lanka. We have now mentioned it as a limitation of the study (line number 456-458). In the Discussion section, we have made a conscious effort to compare each of the key findings of the study with similar studies from the low-and-middle income countries.

Many abbreviations are used in the manuscript. If the journal's editorial policy permits, description of the relationship between abbreviations and abbreviations in one place would be helpful for readers.

Response: Thank you for this important suggestion. We have now prepared a list of abbreviations and have attached this list in the annexure file. If there is a provision, the journal can include the abbreviation list in the main narrative of the manuscript. We will suggest this to the editor for their consideration. 

Round 2

Reviewer 1 Report

The authors have considered the commentaries about improving the method section (describing in-depth interviews) and the results section (by deleting table 1 and reshaping table 2).

Nevertheless, I am still not convinced by the mixed-method design they used. As I said in my previous review, this is a well-conducted qualitative study, but the quantitative results do not bring anything as the sampling is too small (n=9).

The authors answered: “These articles will be used for advocating for better implementation of PSSP among policy makers of the Ministry of Health in Sri Lanka.” In my opinion, it is not the role of a research paper (and it may interrogate some conflicts of interest from the authors vis-a-vis MoH of Sri Lanka).

Authors point out that they used Caffrey et al. (2017) explanatory design, i.e., “how do the qualitative data explain the quantitative results?”. A mixed method design is about the use of quantitative and qualitative data, and quantitative data is about the representativity of the phenomenon studied. In the present study n=9  is probably not representative (i.e., 14% of the PMCI, as pointed out by the authors).

I, therefore, always recommend rejecting the manuscript as it is and only considering the publication of the qualitative study, which is, again, of high quality.

It is now an editorial decision to accept the paper for publication in its present form.

Author Response

The authors have considered the commentaries about improving the method section (describing in-depth interviews) and the results section (by deleting table 1 and reshaping table 2).

Nevertheless, I am still not convinced by the mixed-method design they used. As I said in my previous review, this is a well-conducted qualitative study, but the quantitative results do not bring anything as the sampling is too small (n=9).

The authors answered: “These articles will be used for advocating for better implementation of PSSP among policy makers of the Ministry of Health in Sri Lanka.” In my opinion, it is not the role of a research paper (and it may interrogate some conflicts of interest from the authors vis-a-vis MoH of Sri Lanka).

Response: Thank you for the comment. As it is an operational research, we strongly believe that a published manuscript can be a potent tool for action and therefore has a purpose beyond just generating evidence. In the past we have used similar manuscripts for active advocacy and bringing about changes in policy and practice . As detailed in the previous round of revision, this is one of the three reasons why we feel the the quantitative component is relevant and should be retained in the manuscript.

Authors point out that they used Caffrey et al. (2017) explanatory design, i.e., “how do the qualitative data explain the quantitative results?”. A mixed method design is about the use of quantitative and qualitative data, and quantitative data is about the representativity of the phenomenon studied. In the present study n=9  is probably not representative (i.e., 14% of the PMCI, as pointed out by the authors).

I, therefore, always recommend rejecting the manuscript as it is and only considering the publication of the qualitative study, which is, again, of high quality.

It is now an editorial decision to accept the paper for publication in its present form.

Response: Thank you for the comment. As detailed in the previous round of revision, we are not trying to extrapolate the quantitative study results and accept the limitations with the sample size wholeheartedly. However, we strongly believe the quantitative results are required in the manuscript as it complements the qualitative results. The qualitative results of this study cannot be interpreted in isolation. Therefore, we wish to retain the quantitative results.  

We understand that the reviewer's opinion is to reject the article as we are not able to make any further changes in the manuscript. We shall await and abide by the editor's decision on this matter.